# Structural Insight into La_0.5_Ca_0.5_Mn_0.5_Co_0.5_O_3_ Decomposition in the Methane Combustion Process

**DOI:** 10.3390/nano11092283

**Published:** 2021-09-02

**Authors:** Olga Nikolaeva, Aleksandr Kapishnikov, Evgeny Gerasimov

**Affiliations:** Boreskov Institute of Catalysis SB RAS, 630090 Novosibirsk, Russia; ribka-99@mail.ru (O.N.); avl97@mail.ru (A.K.)

**Keywords:** in situ XRD, phase transformation, perovskite, microstructure, recrystallization

## Abstract

Perovskite-like solid solution La_0.5_Ca_0.5_Mn_0.5_Co_0.5_O_3_ was tested during the total methane combustion reaction. During the reaction, there is a noticeable decrease in methane conversion, the rate of catalyst deactivation increasing with an increase in temperature. The in situ XRD and HRTEM methods show that the observed deactivation occurs as a result of the segregation of calcite and cobalt oxide particles on the perovskite surface. According to the TGA, the observed drop in catalytic activity is also associated with a large loss of oxygen from the perovskite structure.

## 1. Introduction

Among the fossil energy resources, natural gas presents a particular interest because of its higher energy content than coal and petroleum as well as its reduced CO_2_ emissions (50% less than coal and 30% less than petroleum) [1]. The most common catalysts for the total oxidation reactions are the systems based on noble metals (Pt, Pd, and Au) [2,3,4,5]. Despite their high activity, this class of catalysts has a number of disadvantages, namely the high cost and limited reserves of the active component, low thermal stability, etc. The most significant problem is the search for alternative low-cost compounds with high catalytic activity. These requirements are satisfied by solid solutions with the perovskite structure [6,7,8,9]. 

Solid solutions with the perovskite structure have a general formula of ABO_3_, where position A is usually occupied by cations of alkaline earth metals and position B is metals from 3D groups. Despite a simple chemical formula, a huge number of different compounds have been created based on the structure of perovskite [10,11,12]. Variations in the elemental composition and the degree of doping lead to the formation of various physical and chemical properties [13,14,15,16,17,18,19]. 

Currently, a large number of publications are devoted to the optimization of the structure, synthesis methods, and subsequent treatments to improve the reactivity of solid solutions with the perovskite structure. The use of different synthesis techniques allows researchers to obtain solid solutions with homogeneous and heterogeneous structures, and to vary the morphology and size of the particles [10,20,21]. The main purpose of these studies is to obtain a material in which a high thermal stability, the ability to change the charging state, and a high transport capacity of oxygen will be realized [22,23]. The combined application of noble metals and the perovskite structure can also positively affect the catalytic activity. This approach is implemented in the manufacturing of supported components for three-way catalysts or methane dry reforming [24,25,26].

The increase in catalytic activity can be achieved by optimizing the structure of perovskite [27,28]. In the methane oxidation reaction, the catalytic activity was parallel to the reducibility of Mn (IV) in various catalysts [29,30,31]. An increase in the content of Mn (IV) cations can be achieved by doping perovskite with divalent cations. For example, introducing Ca into the La sublattice may increase the catalytic activity, but it may also reduce the thermal stability. In [32], a nonlinear increase in catalytic activity was observed with the introduction of Ca into the perovskite structure. However, [33] shows that calcium cations migrate to the surface as a result of the reaction and can negatively affect catalytic properties. 

The aim of this work was to study the deactivation process of La_0.5_Ca_0.5_Mn_0.5_Co_0.5_O_3_ during the methane combustion reaction via the in situ XRD method. The data obtained can be important for understanding the processes that occur during the reaction and affect not only the surface, but also the catalyst volume.

## 2. Materials and Methods 

La_0.5_Ca_0.5_Mn_0.5_Co_0.5_O_3_ sample was synthesized by the polymerizable precursor method. Appropriate amounts of crystal hydrates of salts, including La(NO_3_)_3_·6H_2_O (ZRM, >99%), Ca(NO_3_)_2_·4H_2_O (Merck, >99%), Mn(NO_3_)_2_·4H_2_O (Merck, >99%), Co(NO_3_)_2_·6H_2_O (Merck, > 99%), citric acid (ChimProm, >99.5%), ethylene glycol (ChimProm, >99.5%), and distilled water were mixed. An aqueous solution with cation ratio La:Ca:Mn:Co of 1:1:1:1 was prepared. The resulting reagent was evaporated at 70–80 °C until the formation of a resinous polymer. The precursor was calcined at 800 °C for four hours with a rise in temperature of 100 °C/h. 

Methane complete oxidation reaction was carried out in a tubular quartz plug flow reactor using a feed of 1% CH_4_ + 10% O_2_ + N_2_ balance at 400–600 °C and a contact time of 0.1 ms. The weight of the powder fraction of 0.25–0.5 mm was 50 mg. Analysis of the reaction mixture was carried out using a gas analyzer equipped with IR sensors for CO, CO_2_, and CH_4_ (Boner LLC, Novosibirsk, Russia). 

Electron microscopy investigation (HRTEM) was performed using a JEM-2200FS (JEOL, Tokyo, Japan) electron microscope operated at 200 kV with a lattice resolution of 0.1 nm.

XRD patterns were obtained on a Bruker AXS D8 Advance diffractometer (Karlsruhe, Germany) equipped with a high temperature, supply of various gas mixtures, the use of CuKα radiation in scanning with a step of 2θ = 0.05° point by point, and an accumulation time of 3 s at each point in a range of the angles 2θ = 15–75°. Temperature measurements were performed according to the following conditions: temperature rate of 10 °C/min with 1% CH_4_ + 10% O_2_ + He flow and a mixture rate of 40 mL/min at 600 °C. Before the measurements, the sample was kept for ~30 min at 450 °C in air conditions. After heating the sample in an air atmosphere, a reaction mixture was fed into the diffraction chamber and heated to 600 °C. Diffractograms were obtained at temperatures of 30, 450, and 600 °C. After cooling to room temperature, a diffractogram was also obtained under the conditions of a reaction medium. Heating in a helium current and an air medium was carried out at a temperature of 600 °C at a heating rate of 10 °C/min. The crystallite sizes and chemical compositions were calculated in the X’Pert HighScore Plus (PANalytical B.V., Almelo, The Netherlands) software. The calculation and refinement of lattice parameters were performed in the IK (BIC SB RAN, Novosibirsk, Russia) software by the method of least squares.

Thermal analysis of the sample was performed using a synchronous thermal analysis device, STA 449C Jupiter (NETZSCH, Selb, Germany). This device combines the methods of differential thermal analysis (DTA) and thermogravimetric analysis (TGA) into one dimension. The weight of the sample was approximately 100 mg. The furnace temperature was increased from 40 to 900 °C at a rate of 10 °C/min with He flux of 30 mL/min. The sample weight was monitored continuously as a function of temperature.

## 3. Results

Figure 1 shows the dependence of methane conversion on temperature and time. As can be seen from the picture, the methane conversion begins to decrease during exposure to one temperature. At 450 °C the conversion drop in 30 min is insignificant and is about 3%, at 550 °C the conversion drop in the same time is about 12%. A further increase in temperature to 600 °C leads to a 20% drop in conversion. A stepwise temperature increase leads to a rise in the methane conversion, but over time partial decomposition of the sample occurs. A rise in the temperature of the reaction medium leads to an increase in the deactivation rate of the catalyst. The subsequent lowering of the temperature in the reactor to 500°C leads to stabilization of the reaction rate at a lower point than in the previous level. It is important to note that the contact time was chosen specifically to be short, because with a longer contact time this perovskite showed almost 100% conversion. In that case, it becomes difficult to observe the process of deactivation of the catalyst.

To study the process of sample deactivation, high temperature XRD experiments were performed simulating the reaction conditions in the diffractometer chamber. Figure 2 shows XRD patterns of the sample in the different states. According to XRD data, the sample in the initial state was a single-phase perovskite of orthorhombic modification (sp.gr. Pnma, JCPDS # 46-513) [34]. When the reaction mixture was fed into the diffractometer chamber, a jump-like increase in the volume of a part of the perovskite occurred with the formation of a phase with cell parameters close to LaMnO_3_ (sp.gr. Pnma, JCPDS # 87-2012) and the simultaneous segregation of CaCO_3_ (sp.gr. R-3c, JCPDS # 5-586), CoO (sp.gr. Fm-3m, JCPDS # 48-1719), and Co_3_O_4_ (sp.gr. Fd-3m, JCPDS # 42-1467) particles on the surface of the perovskite phase. It is important to note that the phase observed during the reaction is close in lattice parameters to LaMnO_3_, but it may contain calcium and cobalt cations in its composition. A selected fragment of Figure 2 shows the difference between the initial and processed perovskite, showing a significant shift of the 200 reflex on the XRD pattern (2θ ~ 33°). Heating to a temperature of 600 °C leads to a significant broadening of reflexes, especially in the area of distant angles. The intensity of reflexes in the areas of 2θ = 50–70° significantly decreased. It can be assumed that the initial effect of the reaction medium on the perovskite structure leads to the decomposition of the solid solution and the formation of a new intermediate structure.

The subsequent treatment of perovskite by the reaction medium leads to a partial return to the initial state (blue XRD pattern). At the same time, the phases of calcite and cobalt oxide are preserved. The phase compositions obtained from the experiment data are shown in Table 1. Further exposure to the reaction medium leads to relaxation of the perovskite structure and gradual recrystallization. The size of the crystallites in the initial state was 270 Å; with the initial impact of the reaction medium, a sharp jump occurred with a decrease in the crystal size to 170 Å. Further heating of the sample in the reaction medium did not lead to changes in the crystal parameters, which indicates the stabilization of the structure. 

The appearance and subsequent disappearance of the intermediate phase of perovskite can be associated with the active release of oxygen from the volume of the structure. The lack of oxygen in the perovskite structure leads to the appearance of extensive planar vacancies (Appendix A) and the migration of cobalt and calcium cations to the surface of the perovskite phase. Similar effects were observed in the LCMFO and LCFO systems [33,35]. According to the experimental data, it can be assumed that with further exposure to the reaction medium, the perovskite structure relaxes and a part of the solid solution returns to a state close to the initial one. However, oxides or carbonates of calcium and cobalt remain on the surface of the perovskite phase. The release of manganese or lanthanum cations to the surface is unlikely in this case, since these types of cations have a larger ionic radius or are more valently stable. Considering these factors, it can be assumed that the formation of an intermediate structure similar to the LaMnO_3_ crystal structure occurred due to the intensive release of calcium and cobalt cations on the surface. In the case of LCCO systems, high proportions of calcium also affect the stability of the solid solutions negatively [36]. Their catalytic activity decreases with an increase in calcium content due to the segregation of heterogeneous particles on the surface [37].

During cooling, the parameters of the unit cell didn’t return to their original state, since additional phases were segregated from the perovskite structure. An interesting feature is the disappearance of the peak in the area of angles of 2θ ~ 42–44°. During heating in the reaction, medium cobalt oxide reflexes appeared on the surface of perovskite, but they partially disappeared after the cooling process, which may indicate a partial embedding of cobalt cations back into the perovskite structure.

To separate the influence of the reaction medium and temperature factors, XRD experiments were conducted on the heating of perovskite in the He current and in the air atmosphere. In the case of the calcination of La_0.5_Ca_0.5_Mn_0.5_Co_0.5_O_3_ in atmospheric oxygen at 600 °C (yellow XRD pattern), no visible decomposition occurred and the perovskite structure was retained quite well (Table 2). 

However, in the case of the experiment that was carried out in a He medium (pink XRD pattern), a jump-like increase in the cell parameters (Table 2) was observed for the perovskite structure, as well as a slight asymmetry in some hkl reflexes (such as 321). These effects can be associated, in particular, with the influence of low oxygen partial pressure in the medium. An increase in the cell parameters and a decrease in the crystallite size to 150 Å can be caused by lattice oxygen loss and the formation of defects in the structure, as was observed in [38].

In situ XRD experiments are in good agreement with the HRTEM data (Figure 3). In the initial state, the sample was a single-phase solid solution with a perovskite structure. Particles with sizes of 20–50 nm formed agglomerates with a micron size. After the reaction, a layer of calcium carbonate was formed on the surface of the perovskite particles, covering the surface of the perovskite. The thickness of this coating varied from a few nanometers to individual agglomerates with dimensions of 10–30 nm. The electron microscopy images show calcium oxide particles; this effect occurs due to the interaction of an electron beam with the structure of calcium carbonate, which leads to the decomposition of the initial carbonate to the oxide state. Particles with the structure of Co_3_O_4_ and sizes of 5–20 nm were also found on the surface of perovskite. It is worth noting that the number of cobalt oxide particles is small, which corresponds to the XRD data.

The TG (Figure 4) curve shows that the sample loses mass in three stages during heating in He flow. The first 1.5% mass change occurs in the range from 50 to 480 °C and can be associated with the desorption processes from the sample surface. From 480 to 650 °C, there is a rapid mass loss of −0.8%. As follows from the DTG curve, the highest rate of mass loss occurs in the region of 600 °C. In the temperature range of 650 to 900 °C, the sample mass decreases by 0.3%.

Taking into account the catalytic measurements and thermogravimetry data, it can be assumed that oxygen is released from the perovskite structure in the temperature range of 500–650 °C. On the one hand, this leads to a short-term increase in catalytic activity, but contributes to the formation of calcium carbonate on the surface of the active phase on the other, which further leads to a decrease in catalytic activity. The stabilization of the perovskite structure under the conditions of the reaction medium occurs through the partial decomposition of the solid solution, with the appearance of the intermediate phase similar to LaMnO_3_ and the further formation of a heterogeneous structure. Indeed, according to the XRD data, the size of the crystallinity decreases significantly, since the release of calcium and cobalt cations on the surface leads to the formation of defects inside the perovskite and, as a result, to a decrease in the size of the perovskite crystals.

## 4. Conclusions

In this work, the reaction medium’s influence on the crystal structure of La_0.5_Ca_0.5_Mn_0.5_Co_0.5_O_3_ was studied by high temperature XRD and HRTEM. The process of complete methane oxidation leads to partial decomposition of the solid solution, which results in the segregation of CaCO_3_ and CoO_x_ particles on the surface of the perovskite phase. The influence of the reaction mixture on perovskite leads to the formation of an intermediate phase with the structure of LaMnO_3_. The process is partially reversible because during the reaction La_0.5_Ca_0.5_Mn_0.5_Co_0.5_O_3_ returns to the initial structural modification, with a change in the parameters caused by the partial removal of cations from the structure. The TGA method shows an intensive mass loss of perovskite in the temperature range of 480–600 °C. Thus, the intense release of oxygen from the perovskite structure leads to the partial decomposition of the perovskite structure and can affect its catalytic properties. It should be assumed that a partial decrease in the proportion of calcium cations in the perovskite structure could have a beneficial effect on thermal stability, since the amount of calcium carbonate on the catalyst surface may decrease.

## Figures and Tables

**Figure 1 nanomaterials-11-02283-f001:**
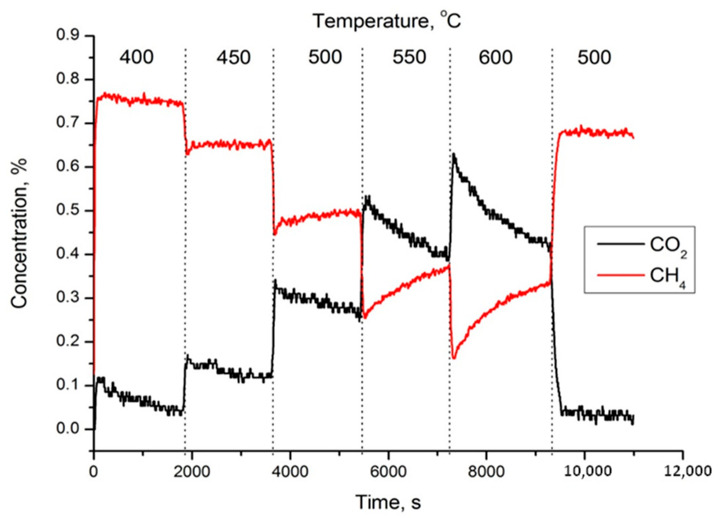
Dependence of the degree of methane conversion on temperature and time.

**Figure 2 nanomaterials-11-02283-f002:**
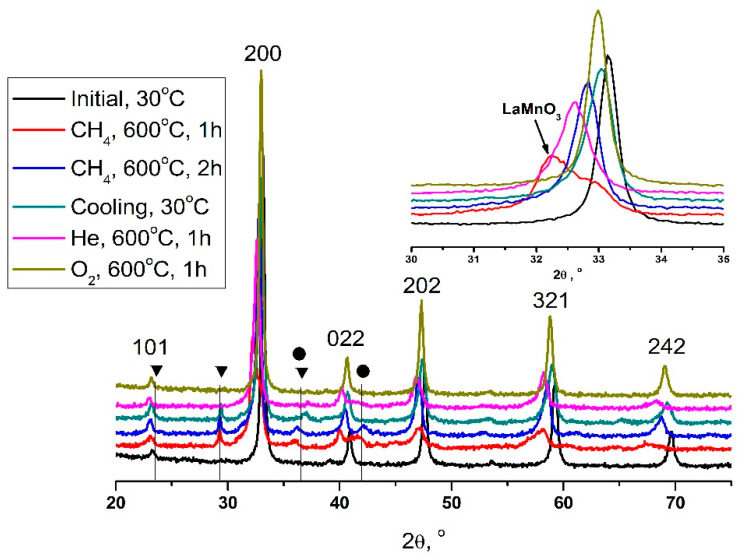
XRD pattern of La_0.5_Ca_0.5_Mn_0.5_Co_0.5_O_3_ obtained under the reaction conditions. The triangles indicate the calcite phase and the cobalt oxide phase is indicated by the circles.

**Figure 3 nanomaterials-11-02283-f003:**
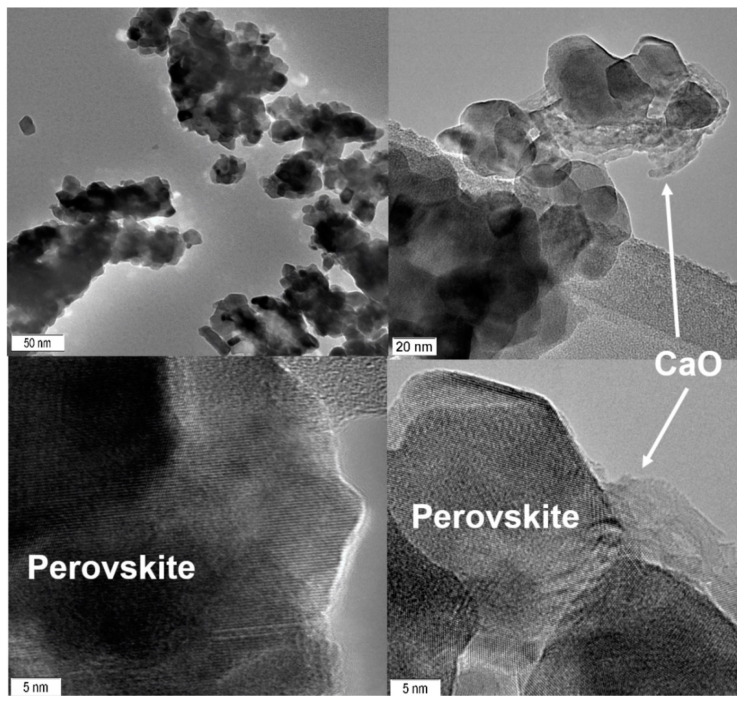
HRTEM images of the sample before and after the reaction.

**Figure 4 nanomaterials-11-02283-f004:**
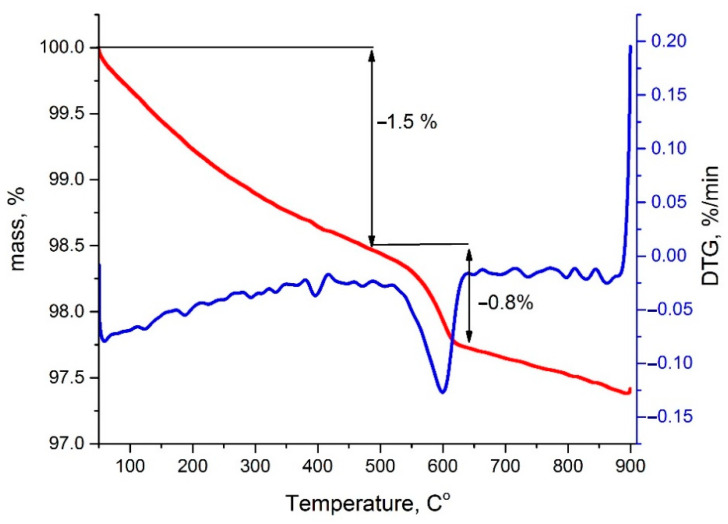
Thermogravimetric curve of La_0.5_Ca_0.5_Mn_0.5_Co_0.5_O_3_ in the He flow.

**Table 1 nanomaterials-11-02283-t001:** Phase composition for La_0.5_Ca_0.5_Mn_0.5_Co_0.5_O_3_ in different thermal treatments (weight fraction %).

Temperature, °C	Perovskite	CoO-Co_3_O_4_	Calcite
25 (initial)	100	0	0
600 (1 h)	84.5	5.9	9.6
600 (2 h)	90.3	2.1	7.7
25 (cooling)	91.1	1.5 (Co_3_O_4_)	7.4

**Table 2 nanomaterials-11-02283-t002:** Lattice parameters and the degree of crystallinity for La_0.5_Ca_0.5_Mn_0.5_Co_0.5_O_3_.

Temperature, °C	a, Å	b, Å	c, Å	Cr. Size, Å
25 (initial)	5.391(1)	7.616(3)	5.419(3)	270
600 (1 h)	5.467(6)	8.030(9)	5.483(7)	120
600 (2 h)	5.453(5)	7.704(5)	5.515(2)	170
25 (cooling)	5.478(4)	7.652(3)	5.412(6)	220
600 (He treatment)	5.478(3)	7.773(7)	5.503(6)	150
600 (O_2_ treatment)	5.438(2)	7.691(2)	5.436(4)	260

## Data Availability

Not Applicable.

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
