# Peer review of "Structural Insight into La0.5Ca0.5Mn0.5Co0.5O3 Decomposition in the Methane Combustion Process"

_nanomaterials, 2021, doi:10.3390/nano11092283_

Round 1
Reviewer 1 Report
Authors present an in-operando X-ray diffraction study of La0.5Ca0.5Mn0.5Co0.5O3 catalyst. Together with a diminution of the catalytic power, they detect a phase decomposition. Paper seems sound, but there are some points that authors should revise before I could consider it suitable for publication.
-It seems (both in the abstract and in the text) that XRD shows that CaCO4 and CoO is formed at the surface of the perovskite phase. XRD cannot give that information, it comes from electron microscopy. In fact, XRD shows that, at least CaCO4 is formed as quite coherent and large crystals.
- Authors claim that La0.5Ca0.5Mn0.5Co0.5O3 decomposes in LaMnO3 CaCO4 and CoO (or nearly on this). This is a very hard statement. X-rays certainly show the formation of CaCO4 and CoO, but some kind of quantative analysis from X-ray data must be done to support this point. Authors have analyzed data (they extract lattice parameters from the different patterns). Can they show any kind of Rietveld refinement of the patterns with the three phases showing which is the amount of Ca/Co truly segregated? Otherwise, the main result of the paper is not fully supported by data.
- What does exactly mean "Planar vacancies"? Do authors mean the concentration of vacancies in a certain plane? How do they arrive to this conclusion.
Reviewer 2 Report
Dear Authors! Thanks for your Communication. But I have the following comments:
- It is wishable to add the references about the mechanism of catalytic fuel combustion on LaMnO3 an its analogues.
- It is neсessary to add the reagents analytical grade.
- Fig. 1. The curves at 400 oC differ from the curves at other temperatures (CH4 content does not rise). What is the reason of this fact?
- Would you please to add a table with phase composition of the samples according to Fig. 2?
- What was the method for estimation of the crystallite size?
- It is neсessary to clarify why did the cell parameters increase with the oxygen release.
- Table 1. Typo in the 4th column: not b.. c. Please add errors to the cell parameters. What was a method for the cell parameters calculation?
- Usually the thermogravimetric experiment is carried out in two stages: the preliminary annealing is necessary for the desorbtion processes and horizontal plateau at low temperatures providing (Fig. 4), isn it?
- Could you please to determine the absolute oxygen content in the sample?
- Would you please to provide the XRD analysis on the sample after TGA?
- Could you please to write down the chemical equation for the samples' decomposition (may be schematically) and the defect equation for the oxygen release illustration.
- It is wishable to make the conclusion about the potential use of the investigated sample in comparison with analogues.
Round 2
Reviewer 1 Report
Authors have made some improvements over the first version that will help readers to better judge the validity of the conclusions. I have only two minor points to add.
First: As it is written (at different points in the paper, e.g. the abstract), it seems that X-rays allow authors to conclude that impurities form at the surface. This is somehow misleading as it is microscopy that gives this information. This must be corrected.
The second is that authors must give some more information on the composition given in table 1: i) Are the numbers the weight fraction? and ii) (and more important) how have they extracted the composition given? Was it by a Rietveld refinement? Which phases have they used? Which La/Ca balance is assumed for the perovskite?
Reviewer 2 Report
Dear Authors! Thanks for your attention to my comments. I have minor corrections else:
- Please clarify the contents of citric acid and ethylene glycol in precursor solution (Part 2).
- Please add the samples' decomposition equations in the CH4 and He atmospheres both to the text for better illustration (or to Supplementary Material) .
- Please add the XRD pattern of the sample after TGA to Supplementary Material.
- New conclusion "It should be assumed that a partial decrease in the proportion of calcium cations in the perovskite structure could have a beneficial effect on thermal stability, since the amount of calcium carbonate on the catalyst surface may decrease" needs in approvement. Please provide the literature or your previous data.
